# A Proactive Recognition System for Detecting Commercial Vehicle Driver’s Distracted Behavior

**DOI:** 10.3390/s22062373

**Published:** 2022-03-19

**Authors:** Xintong Yan, Jie He, Guanhe Wu, Changjian Zhang, Chenwei Wang

**Affiliations:** School of Transportation, Southeast University, Nanjing 210018, China; 230198699@seu.edu.cn (X.Y.); 220173028@seu.edu.cn (G.W.); 230189680@seu.edu.cn (C.Z.); 230208824@seu.edu.cn (C.W.)

**Keywords:** driver’s distracted behavior, proactive recognition system, deep learning approaches, commercial vehicle surveillance system

## Abstract

Road traffic accidents regarding commercial vehicles have been demonstrated as an important culprit restricting the steady development of the social economy, which are closely related to the distracted behavior of drivers. However, the existing driver’s distracted behavior surveillance systems for monitoring and preventing the distracted behavior of drivers still have some shortcomings such as fewer recognition objects and scenarios. This study aims to provide a more comprehensive methodological framework to demonstrate the significance of enlarging the recognition objects, scenarios and types of the existing driver’s distracted behavior recognition systems. The driver’s posture characteristics were primarily analyzed to provide the basis of the subsequent modeling. Five CNN sub-models were established for different posture categories and to improve the efficiency of recognition, accompanied by a holistic multi-cascaded CNN framework. To suggest the best model, image data sets of commercial vehicle driver postures including 117,410 daytime images and 60,480 night images were trained and tested. The findings demonstrate that compared to the non-cascaded models, both daytime and night cascaded models show better performance. Besides, the night models exhibit worse accuracy and better speed relative to their daytime model counterparts for both non-cascaded and cascaded models. This study could be used to develop countermeasures to improve driver safety and provide helpful information for the design of the driver’s real-time monitoring and warning system as well as the automatic driving system. Future research could be implemented to combine the vehicle state parameters with the driver’s microscopic behavior to establish a more comprehensive proactive surveillance system.

## 1. Introduction

In recent years, road traffic accidents have been demonstrated as an important culprit restricting the steady development of the social economy, especially those accidents relating to commercial vehicles. The World Health Organization’s Global Status Report on Road Safety 2018 pointed out that road traffic accidents are the main cause of death for the global population, with about 3700 people dying every day in the world due to road traffic collisions [1]. According to the China Statistical Yearbook, in 2018, a total of 216,200 motor vehicle accidents occurred in China and resulted in 581,000 deaths, 227,400 injuries, with direct property losses reaching 1.31 billion yuan [2]. The survey illustrates that the proportion of deaths and injuries caused by commercial vehicle accidents is much higher than other types of accidents [3].

The occurrence of traffic accidents is closely related to the bad behavior of drivers. Researchers generally believe that about 80% of road traffic accidents were associated with driver factors [4,5,6], such as furious driving, fatigue driving, and distracted driving.

Notably, driver behavior analysis has obtained many of the theoretical achievements in previous studies [7,8,9,10,11,12]. However, one element that has not been studied adequately is the dynamic real-time monitoring of the driver behavior under the existing supervision methods of the networked joint control of the commercial vehicles, since the existing technology mainly acquires dynamic indicators through secret visits by management personnel or the manual viewing of background videos, incapable of automatically identifying the wrong driving posture or behavior and thus providing real-time warning. Besides, the existing mature driver real-time monitoring systems are mostly fatigue-state detection products, which have not been widely promoted to detect the driver’s distracted posture and behavior.

Note that the rise of artificial intelligence (AI) and computer vision (CV) has provided the basis of addressing the aforementioned problems and laid the technical foundation for automatically identifying wrong driving behaviors. Various machine learning and deep learning methods can be more commonly applied. Research results of CV, such as target detection, motion tracking and image recognition classification, are constantly emerging. The computer processing performance is close to the human level and the results on specific data sets even exceed the human level, which provides technical feasibility for the automatic recognition of driving postures.

On the other hand, driver’s posture recognition systems are now mostly developed by automobile manufacturers and used for commercial purposes. At the level of safe driving, drivers of commercial vehicles have greater safety requirements for posture monitoring. However, there is insufficient research on drivers of commercial vehicles. Past work on driver’s posture detection and recognition is mainly aimed at car drivers, whereas relevant studies relating to commercial vehicle drivers are still lacking, especially with regard to truck drivers, a vulnerable group at greater risk of driving.

Additionally, some actual investigations suggest that the commercial vehicle drivers spend about one third of the time driving at night with far worse light conditions relative to their daytime counterparts, and due to the divergent light conditions between daytime and night, there is a substantial difference between the video image data set generated by night monitoring and the daytime monitoring data set. It is noteworthy that posture detection at night is also a promising topic to be investigated, while the previous research focused on posture detection in the daytime, generally with good light conditions.

An alternate limitation of the previous studies regarding the driver’s posture recognition lies in the monotonicity of recognized posture, i.e., single postures were mainly recognized such as normal driving, controlling the gear, calling the phone, and smoking, only covering a small part of the driver’s posture. Multiple postures such as eating or drinking water, using the dashboard, using the mobile phone, not looking ahead and keeping hands off the steering wheel will also greatly affect driving safety, which were not adequately investigated in previous studies. Also, superimposed posture (e.g., the driver uses the mobile phone while controlling the steering wheel) recognition research is relatively blank. Based on the aforementioned issues confronted by the existing driver’s posture recognition systems, there is a great necessity to develop a comprehensive driver’s posture recognition system with multiple recognition objects, scenarios, types, and so on.

In the existing studies, static posture recognition is found to be the major target of the present research relating to driver behavior, which is suitable for the post-recognition of long-period captured images. However, it is inconsistent with the naturalistic driving environment due to its ideal assumption to neglect the dynamism of driver behavior. The intrinsic characteristics including continuity, similarity, superposition and mutual influence of the driver’s posture and behavior have not been studied. To better aid in the enhancement of the current driving safety technique, more research should be conducted to explore the dynamic driver’s posture characteristics using the whole process of video or image sequences.

To contribute to traffic safety analysis regarding driver’s posture recognition systems to prevent distracted driving behavior, the objectives of this research are three-fold: (1) to identify driver’s posture characteristics and provide a methodological framework that demonstrates the significance of enlarging the recognition objects, scenarios and types of the existing driver’s posture recognition in a naturalistic dynamical driving environment; (2) to apply the provided methodological framework to conduct the driver’s posture recognition of commercial vehicles based on both daytime and night models, including multiple distracted driving behavior; and (3) to compare the performances of different models and select the most appropriate recognition model.

The study utilized the cascaded convolutional neural network (CCNN) as the instrument to implement the commercial vehicle drivers’ distracted behavior recognition. The image and video dataset collected from Top Chains International Logistics Company (TCILC) and Hurricane Logistics Company (HLC) were categorized and labeled according to driver’s posture characteristics. Daytime and nighttime models relating to commercial vehicle driver’s distracted behavior recognition were separated and compared. Finally, a holistic model (including both daytime and nighttime) was established with CCNN to determine the accuracy of the recognition and prediction.

## 2. Related Work

### 2.1. Change in Driving Data Acquisition Methods

There are three main methods for obtaining driver’s status and behavior data: vehicle behavior-based detection method, intrusive detection method based on driver physiological parameters and non-intrusive detection method based on computer vision.

The detection method based on vehicle behavior mainly reflects the state and behavior of the driver by detecting the abnormal state of the vehicle. The general detection parameters are the direction, trajectory and speed of the vehicle. This method has a long information transmission path, which leads to information loss. The accuracy of detecting driver status and behavior by this method is relatively low. Qu (2012) established a fatigue recognition model by separating the two states of vehicle active lane change and passive deviation [13]. A real-time monitoring device for the steering wheel developed by an American company showed that if the steering wheel does not move within 4 s, the driver is determined to be in an abnormal state [14].

The intrusive detection method based on the physiological parameters of the driver mainly judges whether it is in a fatigue state by analyzing the changed law of the physiological signal in the driver’s body. Common physiological signals mainly include electroencephalogram (EEG), electrocardiogram (ECG), heart rate and respiratory rate. This method requires the driver to wear sensing equipment such as electrodes, optical marks, and air pressure. The detection method is characterized by high accuracy and is not disturbed by changes in the external environment. However, wearing additional equipment will cause interference to the drivers and cause safety hazards. In addition, the high cost of intrusive detection is not conducive to promotion. Zhao, Zhao, Liu and Zheng (2012) successfully developed a driver fatigue detection system based on brain waves (EEG) and electrocardiogram (ECG) [15]. Jung, Shin and Chung (2014) established a monitoring system for driver health based on electrocardiogram (ECG) [16]. Yu, Sun, Member and Yu (2016) have constructed a remote driver health and fatigue monitoring system based on bio signals such as electrocardiogram (ECG), brain waves (EEG), and eye features [17]. Wang, Wang, Zhang, Wu and Darvas (2014) invented a driver distraction detection system based on brain waves (EEG) [18]. Wang, Jung and Lin (2015) proposed a driver attention tracking system based on brain waves (EEG) [19].

The non-intrusive detection method, based on computer vision, analyzes the driver’s state and behavior through the video or image of the drivers. This method has convenient data acquisition, good safety, and high detection accuracy. It has become the most mainstream method for detecting driver state and behavior. The research methods mentioned later are all computer vision-based detection methods. Wang, Qiong, Wang and Zhao (2010) proposed a driver fatigue detection method based on continuous frames to identify the eye state [20]. Jiménez, Bergasa, Nuevo, Hernández and Daza (2012) proposed a driver attention detection method based on facial posture and gaze state [21].

### 2.2. Expansion of Driver’s Posture Detection Area

In recent years, research on driver posture detection based on computer vision has mainly focused on the driver’s head area, which is mostly used to monitor the driver’s fatigue or distraction. In order to expand from driver state detection to driver behavior detection, it is necessary to extract more information on the driver’s limb. Therefore, some scholars expanded the scope of the study area to the upper body consisting of the head, arms and body.

The driver’s head area research mainly determines the driver’s fatigue and distraction state through the driver’s eye state, gaze direction, face orientation and head posture. Zhang et al. (2013) proposed a driver’s eye location method based on skin color detection and texture features [22]. Fu, Guan, Peli, Liu and Luo (2013) developed a particle filter update parameter method to locate the driver’s head [23]. Zhang et al. (2014) combines the percentage of eyelid closure over the pupil (PERCLOS) algorithm and fuzzy principle to detect driving fatigue [4]. Vicente et al. (2015) proposed a visual system to detect distracted driving [24]. Geng (2018) used infrared images and CCNN to extract the eye area and recognize the open and closed state of the eyes [25]. Yang, Chen and Lei (2018) proposed a driving fatigue detection method based on back projection correction and eye gaze correction [26].

The study of the driver’s upper body area mainly uses the position information of the driver’s head and arms to determine whether the driver’s posture and behavior meet the requirements of safe driving. Zhao, Zhang and He (2013) proposed a method for extracting the driver’s posture features based on skin color, region segmentation and regional centroid distance calculation, using a Bayesian classifier for driver posture recognition [27]. Zhao, Zhang, Zhang and Dang (2014) proposed a driving posture feature extraction method based on contourlet transform and edge direction histogram feature fusion, used multi-classifier integration for driving posture recognition [28]. Gupta, Mangalraj, Agrawal and Kumar (2015) developed an automatic driving assistance system based on skin color recognition and an artificial neural network, which can recognize six postures of the driver [29].

### 2.3. Expansion of Driver’s Posture Detection Application Scenarios

At present, the application scenarios of driving posture detection are gradually expanding from day to night, from simulator to naturalistic driving environments. The ability to complete all-day driver posture detection in real scenes is an important prerequisite for such system products.

Most of the current research focuses on the driver’s posture recognition during the daytime. Some scholars also considered the need for driving safety at night. They used infrared video sensors to detect the posture of drivers at night and extended the application scenarios of driver posture detection from day to night. Kolli, Fasih, Al Machot and Kyamakya (2011) proposed a driver emotion recognition method based on infrared video sensors [30]. Flores, Armingol and La Escalera (2012) successfully developed a driver detection system based on near-infrared video sensors [31]. Kondyli, Sisiopiku and Barmpoutis (2013) proposed a driver motion recognition system based on an infrared depth sensor (Microsoft Kinect) [32]. Cyganek and Gruszczyński (2014) established a driving fatigue detection system based on a near-infrared spectrometer and conventional video sensor [33]. Okuno et al. (2018) used infrared images and depth images to detect the driver’s body posture and face orientation through feature sharing and deep convolutional neural networks [34].

In the case of a lack of naturalistic driving environment data sets in the early research, researchers often used driving simulator data sets for experimental tests. In recent years, scholars also choose simulated driving data for rapid optimization when trying new algorithms to recognize driver’s posture, and then generalize in the naturalistic driving environment data set, so as to realize the expansion from driving simulation scenarios to naturalistic environment scenarios. Eren, Celik and Poyraz, (2007) developed a system based on webcams to monitor the activities of car drivers using simulated driving images [35]. Sekizawa, Inagaki, Suzuki and Hayakawa (2007) used a three-dimensional driving simulator to propose a method of modeling and recognition of human driving behavior based on a random switching autoregressive exogenous (SS-ARX) model [36]. Yamada (2016) proposed a high-precision method for detecting the position of the driver’s joints using depth images and color images. He also proposed an algorithm for detecting abnormal driving postures [37]. Sigg (2018) used a driving simulator and Wi-Fi sensor to complete the multi-label classification task for driving posture [38].

### 2.4. Optimization of Driving’s Posture Detection Technology

With the development of computer vision, the technology of driving posture detection has also made great progress. From the early skin color detection model method, to the mid-term detection method based on machine learning, and then to the widely used deep learning intelligent detection method. The robustness, accuracy and real-time performance of the method have been greatly improved.

Early driver detection technologies mostly used skin color detection models, including RGB model, YCrCb model and HSV model. The skin color model can quickly extract the driver’s head, hands and other feature areas from the image, however, this method is less robust to light. Once the light changes greatly, the recognition accuracy will drop sharply. Yan et al. (2016) uses a Gaussian mixture model to detect skin color regions, and uses deep convolutional neural networks to classify driver’s behavior [39].

The mid-term driver’s posture detection technology began to change to other image processing methods and machine learning methods. The basic idea of the method is to collect a picture of the driver and train a classifier to detect the features of the posture. Common classifiers include support vector machines, decision trees, random forests and shallow neural networks, etc., which solves the problem of skin color models interfered by light. Wu et al. (2018) proposed a driver feature detection method based on the center of mass of both arms, using decision trees and KNN methods to identify the driver’s posture [40].

At present, deep learning technology is developing rapidly. Driver’s state and behavior detection based on deep learning is very extensive. This method mainly uses a deep convolutional neural network to realize image recognition. After training with a large amount of image data, the recognition performance far exceeds the traditional machine learning methods. Hu (2018) proposed a real-time driver head positioning method based on the YOLO algorithm, which used deep convolutional neural networks to extract features to recognize postures [41]. Zhao (2018) proposed a locally integrated convolutional neural network model to detect driver fatigue status through facial multi-source dynamic behavior [42].

## 3. Driver Posture Analysis and Video Data Collection

### 3.1. Analysis of Driver’s Distracted Behavior Characteristics

At present, most research regarding driver’s posture recognition focuses on the investigations and improvement of recognition algorithms with only three to four kinds of static attitude posture data sets, neglecting the exploration of the characteristics of driver’s posture itself. It should be noted that the characteristic analysis of the driver’s posture also has a great influence on posture recognition, and this paper has made some attempts in this aspect. By observing the naturalistic driving samples of commercial vehicle drivers more than 30 h, it can be considered that the driver’s posture has the following significant characteristics.

Continuity

In the traditional driver’s posture data set, the body movements of the driver in the same posture are basically similar, which require less processing and training of the data set. However, in the naturalistic driving environment, the driver’s posture is a continuous behavior and the continuity of the posture has a great influence on the definition of the posture boundary. The dynamic posture needs to be further decomposed considering a better coordination of its continuity and diversity. As far as the dynamic posture of drinking water is concerned, the driver takes the water bottle, opens the cap, and then sends the water to the mouth, drinks water, puts down the bottle, tightens the cap, and puts back the water bottle, which can be considered as a continuous process. However, given the diverse characteristics of these postures, a clear definition should be determined, such as which part can be identified as a posture of drinking water, and which part can be thought of as a posture of taking things.

As shown in Figure 1, the action of holding the bottle above the steering wheel is defined as the posture of taking things. The action of opening the bottle cap and taking the bottle to the mouth and drinking is defined as drinking or eating. Similar postures like eating and drinking, smoking, and taking things all intersect with each other and are more difficult to segment. A similar method of defining boundaries is needed.

2.Diversity

Human behavior tends to be ambiguous and difficult to predict. Although the behavior of the driver will be constrained by the driving vehicle, different people will have different driving habits and styles and even the same person will have many differences in driving posture and behavior. The driver’s postures in the cab can be extensively multiple, composed of the behavior of different body parts, such as the head, torso, and hands. The overall posture of each part will have diverse characteristics, which lead to various postures. In addition, the same posture may also have large differences. For example, in a normal driving posture, there is a big difference between straight-forward driving and cornering driving.

3.Superposition

The overall posture of the driver can be formed by superimposing different postures, which is also one of the leading causes of the diversity of postures. Some behaviors of drivers are not mutually exclusive and there is no obstruction between them, especially the one-handed driving posture, which can be compatible with most postures. For example, gear controlling, smoking and one-handed driving can simultaneously occur. As shown in Figure 2, three driver postures are recognized simultaneously in one image.

4.Similarity

Part of the driver’s postures is similar in the image due to the similarity of their head, torso and hand positions, which will cause errors in the recognition of the algorithm (Figure 3). This situation is generally divided into two types. One is that the difference between the two postures is only in the small objects in the hand, which has little effect on driving behavior. For example, normal driving with two hands is very similar to normal driving with two hands holding a cigarette, and the latter can be considered as normal driving with both hands. The other is that the two postures are similar, as shown in the image, whereas they are actually quite different actions with different effects on driving safety. For example, there is a strong similarity between manipulating the instrument panel and taking things near the instrument panel, which is also marginally similar to using the mobile phone/rack. Furthermore, smoking and eating can sometimes be regarded as a similar posture.

5.Transitional

Due to the continuity of the driver’s posture, after explicitly defining the boundary of the posture, there will be an intermediate area between different postures (as shown in Figure 4), indicating that there exists a transitional interval between postures, which is common in one-hand driving. For example, there is a short one-hand driving transition between two-hand driving and controlling the gears.

6.Mutual influence

There is a mutual influence between postures. One is the mutual exclusion of some driving postures. For example, drinking water and smoking cannot be generated at the same time. The other is the mutual influence of the before–after posture on the transitional posture. The analysis of transitional poses is helpful to integrate the categories of before–after actions, so as to distinguish poses with strong similarity. Besides, further judgements of specific posture can be made to determine which action it continues or transits based on its before–after actions.

### 3.2. Video Data Collection and Label Processing

#### 3.2.1. Video Data Collection

Video data were captured and split regarding commercial vehicle drivers from TCILC and HLC, including 666,060 images (452,840 images during the day and 213,220 images at night). The object of the data sets is mainly truck drivers who are at higher risk, and thus, have higher research value. The data sets were divided into naturalistic driving data sets and supplementary driving data sets. The naturalistic driving data sets consist of naturalistic driving images of the truck drivers, with a relatively low proportion of dangerous postures. To avoid the imbalance of the dataset and ensure the accuracy of the prediction for the dangerous postures, this study established a supplementary driving image dataset to capture more comprehensive samples of dangerous postures, by means of reproducing the dangerous driving postures and actions by the truck drivers in static state.

The data acquisition can be categorized into three stages:(1)Preparation stage, a primary investigation was firstly conducted on the cockpit structure of commercial trucks to learn about the distracted behavior of commercial truck drivers. In this stage, the appropriate view was also selected for obtaining the best shooting effect to reduce the avoidable error that might occur to the model. As shown in Figure 5, through the field test, the No. 3 and No. 5 view points were abandoned as the line of sight could not cover all the key areas, and the No. 4 visual angle point was abandoned as it would affect the driver’s driving. Finally, the No. 1 and No. 2 visual angle points were selected as the best view;(2)Car-following data acquisition stage. In this stage, a total of 10 h of non-interference dual view video data was captured from eight professional commercial truck drivers in the natural driving state;(3)Targeted-data acquisition stage. Due to insufficient video data obtained from car-following data collection to build the database behind, a supplementary shoot was conducted aimed at those with distracted behavior, including eating, smoking, making a phone call, operating the dashboard, etc. In this stage, a 200 min driving video data of eight drivers was collected.

#### 3.2.2. Behavior Labelling

##### Distracted Behavior Identification

By observing all the video images of the driver’s posture data set, it was found that the driver’s posture mainly depends on the position of the head and hands, and secondarily depends on the position of objects that may cause dangerous driving actions, such as headphones, cigarettes, dashboards, mobile phones, etc. Different combinations of head, hands, and object positions constitute the complex categories of driver’s posture. The driver’s posture should be properly regulated and simplified to a certain extent. Generally, it can be categorized from three aspects: the position of the hands; the position of the head; and the relative position of the objects to the hands and head. Four key areas (KA) were determined for identification as presented in Figure 6.

Finally, the distracted behavior labels were set as two-hand driving (DB1), one-hand driving (DB2), no-hand driving (DB3), not looking ahead (DB4), smoking (DB5), calling the phone (DB6), controlling the gear (DB7), using the rack (DB8), using the dashboard (DB9), eating or drinking water (DB10), taking things (DB11).

**KA1**:driver’s head area, which mainly involves actions such as smoking, using headphones, eating;**KA2**:vehicle steering wheel area, involving all actions;**KA3**:vehicle operation panel and mobile navigation display area, including operation navigation and instrument panel;**KA4**:vehicle shift lever area, involving actions mainly including one-hand driving, shifting and handling things.

##### Superposition of Driving Posture

According to the current study, the driver’s posture is supposed to be superimposed. The driver may make multiple postures at the same time. At this time, the result cannot be recognized by the model of single posture. Therefore, it is necessary to analyze which actions can be superimposed, and which actions cannot be superimposed owing to their compatibility and mutual exclusion. Considering the superposition of the driver’s postures, 53 types of driver’s postures were identified, including eight two-hand driving postures and 44 one-hand driving postures. It should be stated that all no-hand driving was classified into one category, denoting extremely dangerous postures.

As shown in Table 1, the definition of two-hand driving (DB1), one-hand driving (DB2) and no-hand driving (DB3) can be obtained by judging the position of the hands. The other two position categories (i.e., the position of the head and the relative position of the objects to the hands and head) can be thought of as action postures. State postures are internal exclusive, for example, one-handed driving and two-hand driving cannot exist at the same time. However, there is external compatibility between state postures and action postures, for example, two-hand driving and not-looking-ahead driving can exist at the same time. Besides, action postures are partially internal compatible, e.g., the driver can be smoking when calling the phone.

##### Label Processing

Data labelling processing mostly uses a dichotomous variable to indicate “yes” and “no”, i.e., “1” means “yes” and “0” means “no”. In the traditional labelling processing method, if there are N kinds of recognition results, a 1 × N array consisting of N dichotomous variables can be used. However, given the superposition of postures, there are as many as 53 types of driver postures. If a 1 × 53 array is used, the calculation will become substantially redundant with low efficiency. Thus, this study applied the Orthogonalization method to simplify the label processing.

The core of the Orthogonalization method is that each independent variable only changes a specific attribute of the dependent variable, i.e., one indicator only affects one attribute, the vertical idea was used to avoid one indicator corresponding to multiple attributes. With the Orthogonalization method, each posture can be regarded as a dichotomous variable to form a 1 × 11 array. Besides, since there can be multiple variables with the value of “1”for the label, the representation of the superimposed posture can be used to simplify the label processing and thus improve the calculation efficiency of the subsequent models.

Thus, driver’s posture label set in this research based on the Orthogonalization method is as follows: [DB1, DB2, DB3, DB4, DB5, DB6, DB7, DB8, DB9, DB10, DB11], each label in the label set can be quantitatively demonstrated by the dichotomous variable, for example, the posture of one-handed driving, smoking and controlling the gear can be expressed as [0 1 0 0 1 0 1 0 0 0 0]. More details about the label can be seen in Appendix A (Table A1).

## 4. Methodology

### 4.1. The Standard Module of Convolutional Neural Network

To realize the real-time requirements of the recognition model, the network depth should be reduced as much as possible on the basis of ensuring that the accuracy is greater than 98%. After testing, this study established a CNN model with three convolution blocks to satisfy the aforementioned need, including three convolutional layers, three pooling layers and one fully connected layer. The dimensions and calculations of the convolution kernels used in each layer of the network can be seen in Table 2 and Table 3. The training process has been presented in Figure 7. Besides, the learning rate and batch size set in the current study is 0.001 and 128 regarding the balance of iterating speed and model accuracy.

### 4.2. Separation of Day and Night Models

The driving environment of commercial vehicle drivers includes a daytime driving environment and a nighttime driving environment. As the two driving environments have very different light conditions, it is unrealistic to collect and process data in the same way. At present, cameras that are used for both daytime and nighttime pictures, are shot on the market and generate color images during the daytime and rely on an infrared imaging technique to generate infrared images at night. It should be emphasized that these two images are quite different.

Each color channel in the RGB color images has different histograms, as shown in Figure 8a. With regard to the infrared image, which should be a single-channel image, this sometimes will be automatically separated into RGB channels with the same histogram by the cameras, as shown in Figure 8b. Intuitively, the grey image is very similar to the infrared image, however the main difference is that the infrared image is an image obtained by focusing on the intensity of infrared light of the object, while the grey image is made by focusing on the intensity of visible light of the object. Therefore, when processing infrared images at night, three channels can be synthesized into one channel to save computing resources.

### 4.3. Multi-Task Classification Model

Owing to the diversity of driving postures, if each posture is transformed into a dichotomous variable, the label set will be represented by a marginally long array of 1 × 11 dimensions. Besides, the value of “1”, which represents a certain posture that has occurred, may take place frequently in the array of the label set due to the superimposition of the driver’s postures. However, the activation function of the corresponding output layer in the CNN model is generally the sigmoid function and the softmax function, both of which can only apply the dichotomous variable to represent the features and labels and will show bad performance when dealing with the model containing numerous features and labels with the value of “1”. Therefore, to address this problem and ameliorate the accuracy of algorithms, multiple sub-models of CNNs were employed in the current research.

According to the compatibility and mutual exclusion of drivers’ postures discussed in the previous part of this study, the long label data set of 1 × 11 dimensions was separated into multiple sub-labels to construct the multi-cascaded CNN sub-models. Two-hand driving, one-hand driving and no-hand driving are mutually exclusive postures and cannot occur at the same time, which were labeled with one-hot coding method [43,44] and combined as the hand posture set. For this posture set, the softmax function was used as the output layer activation function of this CNN sub-model.

The postures of not looking forward, smoking and calling the phone are basically compatible with other postures and were classified as three posture sets. Sigmoid function was used as the output layer activation function of these CNN sub-models. The remaining postures are controlling the gear, using the mobile phone/rack, controlling the dashboard, eating or drinking water, and taking things. With the case of no-handed driving excluded, these five postures are mutually exclusive postures and they are less likely to occur simultaneously, which were also labeled with one-hot method. The five types of postures were categorized as behavior posture sets and the softmax function was used as the output layer activation function of this CNN sub-model.

Therefore, the original long label set of the 1 × 11 array was separated into three sub label set, as shown in Equation (1):Label = (Y_hand_, Y_head_, Y_smoke_, Y_phone_, Y_behaviour_)(1)
where Y_hand_ represents the hand posture label set, including two-hand driving, one-handed driving, and no-handed driving, which belongs to one-hot label and uses the softmax activation function. Y_head_, Y_smoke_, and Y_phone_ are the postures of not looking forward, smoking, and phone calling, respectively. They all belong to the dichotomous label and use the sigmoid activation function. Y_behaviour_ represents the label set of behavior postures, including controlling the gear, using the mobile phone/rack, using the dashboard, eating or drinking water and taking things, which also belongs to the one-hot label and uses the softmax activation function.

In addition, the behavior posture label set Y_behaviour_ is different from the traditional one-hot label. There is only one “1” tag in one-hot tag. Due to the mutual exclusion of behavior postures, Y_behaviour_ only has at most one “1” label, however, there may exist the situation where all labels are “0”, i.e., no behavior posture takes place at this posture set. At this time, the CNN model whose output layer is the softmax activation function cannot accurately recognize all the “0” labels, so it is necessary to construct a standard one-hot label by means of the amplification column.

The Y_behaviour_ label set with M × 5 dimensions can be processed by the amplification column. First, one must construct the augmented column matrix ζ with dimension M × 1, and then search for the presence of the “1” label in the Y_behaviour_ label set in M_i_. If it exists, add “0” to the corresponding M_i_ amplification column matrix ζ. if it does not exist, add “1” to the corresponding M_i_ amplification column matrix ζ. Finally, the Y_behaviour_ label matrix and the amplification column matrix ζ are combined to form a standard one-hot label matrix Y’_behaviour_ with dimension (M, 6). The calculation process is as follows:(2)$$\zeta \left(M_{i}\right)=\{\begin{array}{c} 0,\;\;and\;1\in Y_{\mathrm{behaviour}}\left(M_{i}\right)\\ 1,\;\;and\;1\notin Y_{\mathrm{behaviour}}\left(M_{i}\right) \end{array},i\in (0,M)\;$$
(3)$${Y^{\prime }}_{\mathrm{behaviour}}=(Y_{\mathrm{behaviour}},\;\zeta )$$

### 4.4. Driver Posture Recognition Based on a Cascaded CNN Model

Although the CNN model with five sub-models can enhance the accuracy of the overall driver’s posture recognition model, it has some inevitable shortcomings such as lower processing speed and more occupancy of the calculation resources.

To address the issues triggered by constructing sub-models of CNN separately, a cascaded CNN model was proposed in this study. In this cascaded CNN model, the recognition process was designed and optimized based on the compatibility and mutual exclusion of different postures, as shown in Figure 9. For example, if the posture is recognized as a two-hand driving of the hand posture set, this posture will be regarded as contradicting five postures of the behavior posture set, so only four CNN sub-models (i.e., the hand posture set, not looking ahead, smoking and, calling the phone) will be passed through by the input posture data before the model provides the final outcome. Also, if the posture is recognized as no-handed driving, the cascaded CNN model will directly judge it as an extremely dangerous posture and output the outcome, with no need to pass the other sub-models, which can reduce the time and resource consumption and increase the calculation efficiency of the model.

## 5. Results and Discussion

In the daytime data set, 88,970 images were used as the training set and 28,440 images were used as the test set. With regard to the night data set, 46,120 images were used as the training set and 14,360 images were used as the test set. A cascaded CNN model including five sub-models was established and to assess the performance of the current model, accuracy and training error analyses were conducted based on the five sub-models. Furthermore, recognition results of the overall model were discussed.

### 5.1. Sub-Model of Hand Posture Set

The hand posture set model can recognize the driver’s postures of two-hand driving, one-hand driving and no-hand driving. The accuracy of the daytime model can achieve 100% on the training set after 10 iterations and the average training time of a single image is 168.4 ms. The model training accuracy performance and training errors are shown in Figure 10a. The recognition accuracy rate of the hand posture set model on the test set can reach 99.3%, the test error is only 0.028 and the average test duration of a single image is 91 ms.

The accuracy of the night model can achieve 99.71% on the training set after 10 iterations and the average training time of a single image is 151.7 ms. The model training accuracy rate performance and training error are shown in Figure 10b. The recognition accuracy rate of the hand posture set model on the test set can reach 98.14%, the test error is 0.054 and the average test duration of a single image is 82 ms.

### 5.2. Sub-Model of No Looking Ahead

The model of not looking ahead can identify whether the driver has a dangerous posture of not looking ahead. After 10 iterations of the daytime model, 100% accuracy can be achieved on the training set and the average training time of a single image is 169.6 ms. The model training accuracy performance and training error are shown in Figure 11a. The recognition accuracy rate of this sub model on the test set can reach 98.94%, the test error is only 0.031 and the average test duration of a single image is 93 ms.

The night model can achieve 100% accuracy on the training set after 10 iterations and the average training time of a single image is 152.1 ms. The model training accuracy rate performance and training error are shown in Figure 8b. The recognition accuracy rate of this sub model on the test set can reach 99.78%, the test error is only 0.077 and the average test duration of a single image is 80 ms.

### 5.3. Sub-Model of Smoking

The smoking sub-model can identify whether the driver has a dangerous posture when smoking. Note that the daytime smoking sub-model reached 100% accuracy earlier than 10 iterations on the training set and the average training time of a single image is 171.7 ms. The model training accuracy rate performance and training error are shown in Figure 12a. The recognition accuracy of the smoking sub-model on the test set can reach 98.59%, the test error is 0.046 and the average test duration of a single image is 90 ms.

The night model can also achieve 100% accuracy on the training set before 10 iterations and the average training time of a single image is 152.8 ms. This indicates that the smoking sub-models may be more time-saving. The model training accuracy rate performance and training errors are shown in Figure 12b. The recognition accuracy of the smoking sub-model on the test set can reach 98.6%, the test error is 0.047 and the average test duration of a single image is 79 ms.

### 5.4. Sub-Model of Calling the Phone

The calling the phone sub-model can identify whether the driver has a dangerous posture when making a phone call. Similar to the smoking sub-model, 100% accuracy can be achieved earlier than 10 iterations on the training set and the average training time of a single image is 168.9 ms for the daytime calling-the-phone model. The model training accuracy performance and training error are shown in Figure 13a. The recognition accuracy rate of the phone calling sub model on the test set can reach 99.3%, the test error is only 0.018 and the average test duration of a single image is 89 ms.

The night model can achieve 100% accuracy on the training set before 10 iterations and the average training time of a single image is 153.3 ms, also indicating the time-saving characteristics. The model training accuracy performance and training errors are shown in Figure 13b. The recognition accuracy of the phone calling sub model on the test set can reach 98.6%, the test error is only 0.023 and the average test duration of a single image is 81 ms.

### 5.5. Sub-Model of Behavior Posture Set

The behavior posture set model can identify five dangerous postures of controlling the gear, using the mobile phone/rack, using the dashboard, eating or drinking water and taking things. For the behavior–posture–set daytime model, 100% accuracy can be achieved after the 5th iteration on the training set and the average training time of a single image is 172.6 ms. The model training accuracy performance and training error are shown in Figure 14a. The recognition accuracy rate of the behavior posture set sub-model on the test set can reach 99.35%, the test error is only 0.017 and the average test duration of a single image is 89 ms.

The night model can achieve 100% accuracy on the training set after the sixth iteration and the average training time of a single image is 152 ms. The model training accuracy performance and training errors are shown in Figure 14b. The recognition accuracy rate of the behavior posture set sub-model on the test set can reach 99.77%, the test error is only 0.014 and the average test duration of a single image is 81 ms.

### 5.6. Recognition Results of the Overall Model

The model performance of daytime data set is summarized in Table 4. It can be illustrated that the daytime cascaded CNN model has advantages in recognition accuracy and speed over the non-cascaded CNN model. Although the non-cascaded model has already shown a high level of recognition accuracy up to 97.83%, the recognition accuracy of the cascade model is higher, with an increase of 0.85%, reaching 98.68%. Besides, the recognition speed of the cascaded model is 10.4% faster than that of the non-cascaded model. Some scholars have found that the CNN model tends to have a satisfactory recognition performance to deal with a highly demanding dataset [45,46], and this study has also proved this.

Table 5 presents the model performance of night data set. Similar to the daytime model, the night cascaded CNN model has advantages in recognition accuracy and recognition speed over the non-cascaded CNN model. Relative to the daytime cascaded CNN model, the accuracy rate of daytime cascaded CNN model is marginally lower (97.83% versus 97.48%), while the speed shows an increasing tendency, with the running time decreasing from 452 ms to 403 ms. A possible reason for the higher recognition speed of night model is that compared to the daytime data set, the night data set is smaller in the sample size and only needs to deal with the images with a single color channel. With regard to the lower accuracy of the night model, it might be due to the smaller sample size. An alternative explanation might be that the existing image capture technique at night is still immature, which cannot present more unambiguous details of the night image and leads to a relatively lower accuracy of the night model compared to its daytime model counterpart. However, no matter whether the daytime model or the night model was being used, it was found that the cascaded model presents better performance than the non-cascaded model in terms of accuracy and speed.

## 6. Conclusions

This study aimed to ameliorate the traffic safety from the perspective of driver’s posture recognition and provide a more comprehensive methodological framework to demonstrate the significance of enlarging the recognition objects, scenarios and types of the existing driver’s posture recognition systems. Driver’s posture characteristics were primarily analyzed to provide the basis of the subsequent modeling. Five CNN sub-models for driver’s posture recognition were established for different posture categories and to improve the efficiency of recognition, an overall multi-cascaded CNN framework was proposed. To compare the recognition effect of different models and suggest the best model, an image data set of commercial vehicle driver postures, including 117,410 daytime images and 60,480 night images, was trained and test. The important findings and contributions are as follows:(1)The previous studies did not pay enough attention to driver’s posture characteristics analysis while this paper found that driver’s postures have the characteristics of continuity, diversity, superposition, similarity, transitional, and mutual influence (compatible and exclusive). The analyses of these characteristics can not only facilitate better understanding of driver behavior, but also aid in the improvement of the algorithms of the real-time monitoring systems to enhance the recognition speed and prevent the dangerous driving behavior;(2)Compared to the non-cascaded models, both daytime and night cascaded models show better performance in recognition accuracy and speed. Although the accuracy of sub-models and non-cascaded models has reached the application requirements, the accuracy of cascaded models performs better. It can be illustrated that the speed of the cascaded models is about 10% higher than that of the non-cascaded models. This finding can provide a new insight and aid in the establishment of the driver real-time warning system, to include cascaded CNN models instead of non-cascaded CNN models. Additionally, this finding may also be available for the improvement of automatic driving technology;(3)Night models appear to have worse accuracy and better speed relative to their daytime model counterparts for both non-cascaded and cascaded models. Worse light conditions at night and deficiency in the existing night image shot technique might be leading causes, indicating the necessity to improve the monitoring ability at night of the current driver monitoring systems.

These findings could be used to develop policies and countermeasures to improve driver safety. Furthermore, they can provide helpful information for the design of a driver real-time monitoring and warning system, as well as an automatic driving system. It should be noted that this research is not free of limitations. Future studies could be conducted to further consider the dynamic changes of certain dangerous postures and evaluate different danger levels. Additionally, more datasets could be established or opened with multiple scenes to further develop and ameliorate the current model, such as distracted behaviors that can result in reckless driving or unsafe speeds when the drivers are talking with passengers or texting. Likewise, the combination of vehicle state parameters and driver’ postures can be implemented to establish a more comprehensive evaluation system.

## Figures and Tables

**Figure 1 sensors-22-02373-f001:**
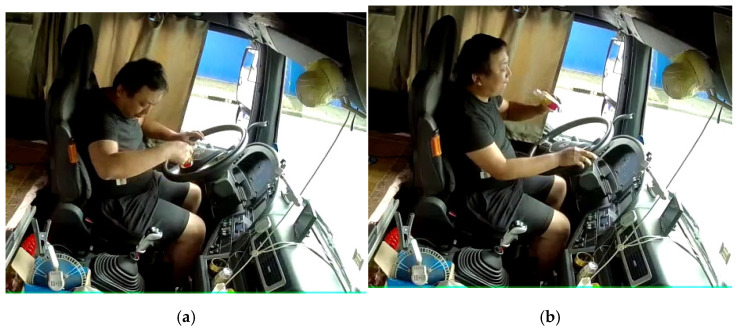
Commercial vehicle driver postures: (**a**) taking things; (**b**) eating or drinking water.

**Figure 2 sensors-22-02373-f002:**
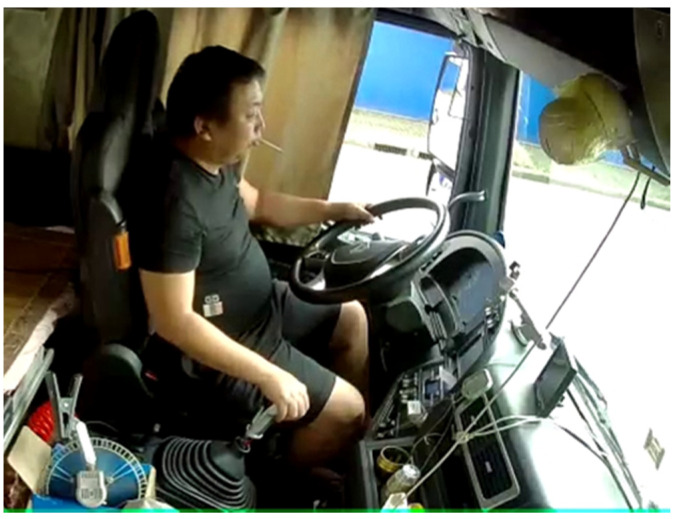
Superimposed posture of one-handed driving, smoking, manipulating gear.

**Figure 3 sensors-22-02373-f003:**
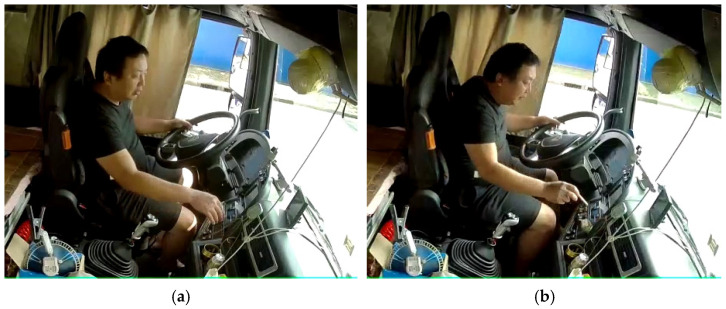
Similarity of commercial vehicle driver postures: (**a**) controlling the dashboard; (**b**) taking things.

**Figure 4 sensors-22-02373-f004:**
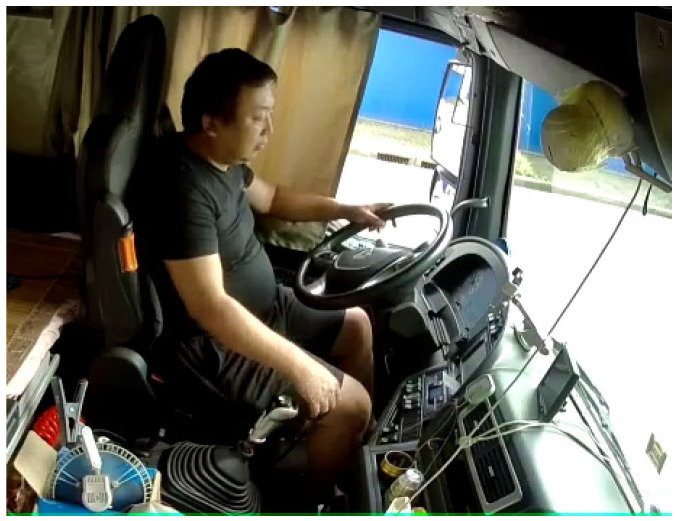
Transitional posture of one-handed driving.

**Figure 5 sensors-22-02373-f005:**
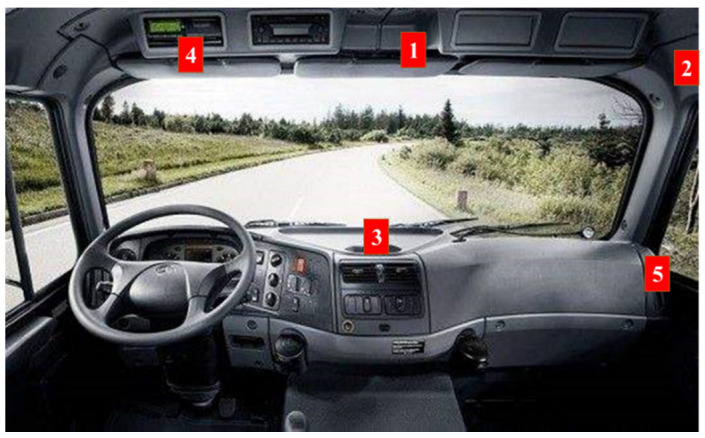
View point selection diagram.

**Figure 6 sensors-22-02373-f006:**
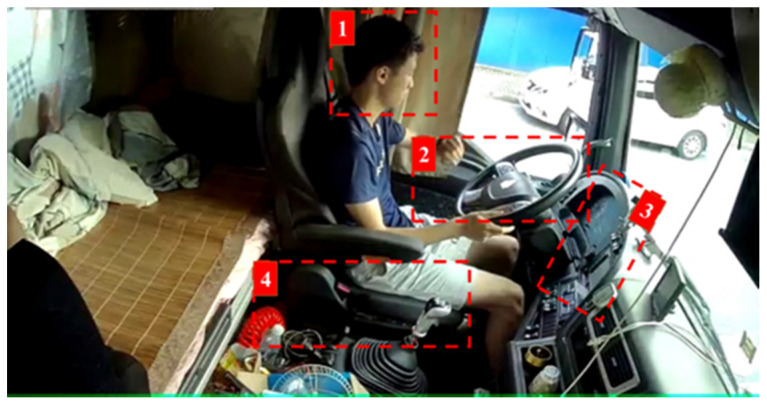
Selection of KA.

**Figure 7 sensors-22-02373-f007:**
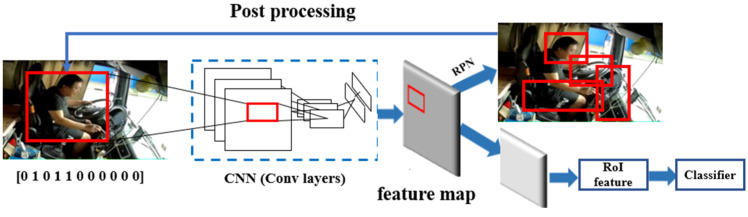
Basic training process of CNN utilized in this study.

**Figure 8 sensors-22-02373-f008:**
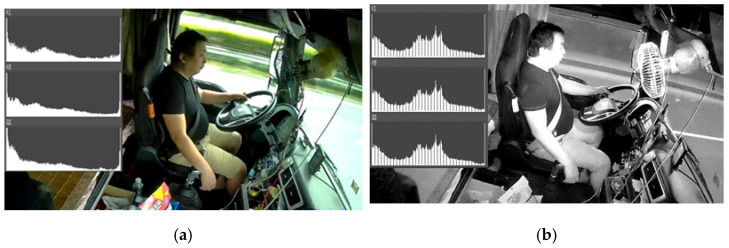
Images and their histogram features: (**a**) daytime image; (**b**) night image.

**Figure 9 sensors-22-02373-f009:**
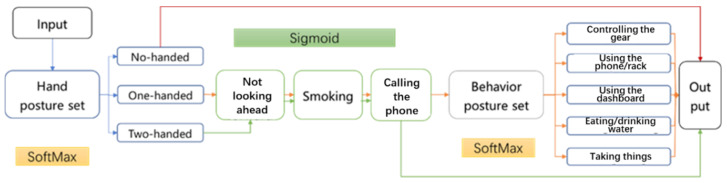
Structure of cascaded CNN models.

**Figure 10 sensors-22-02373-f010:**
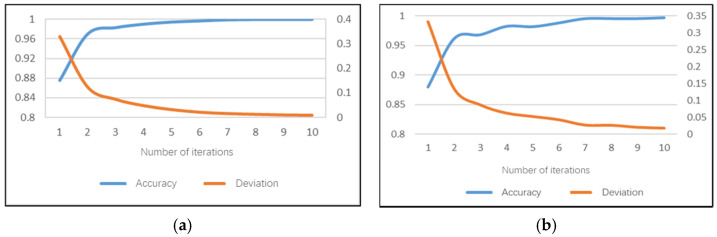
Training performance of hand posture set sub-model: (**a**) daytime; (**b**) night.

**Figure 11 sensors-22-02373-f011:**
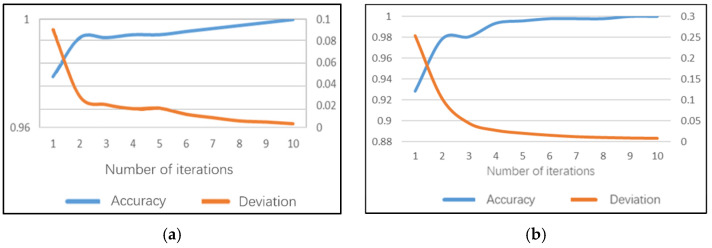
Training performance of not looking ahead sub-model: (**a**) daytime; (**b**) night.

**Figure 12 sensors-22-02373-f012:**
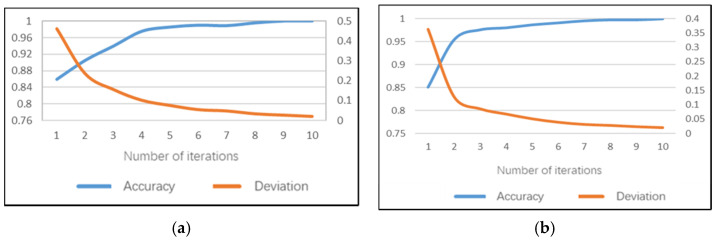
Training performance of smoking sub-model: (**a**) daytime; (**b**) night.

**Figure 13 sensors-22-02373-f013:**
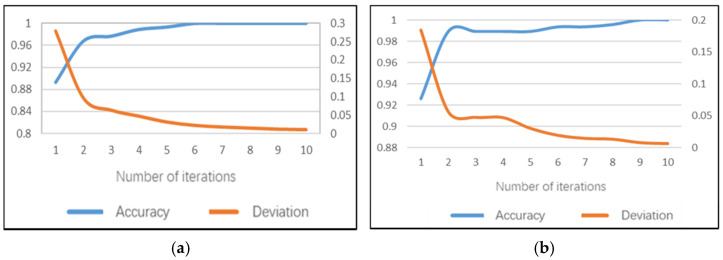
Training performance of phone calling sub-model: (**a**) daytime; (**b**) night.

**Figure 14 sensors-22-02373-f014:**
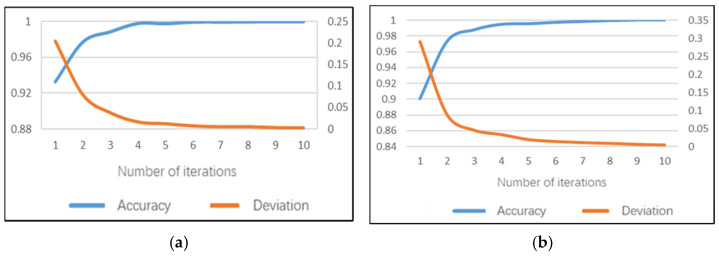
Training performance of behavior posture set sub-model: (**a**) daytime; (**b**) night.

**Table 1 sensors-22-02373-t001:** Compatibility and mutual exclusion of different postures.

	DB1	DB2	DB3	DB4	DB5	DB6	DB7	DB8	DB9	DB10	DB11
**DB1**	—	×	×	√	√	√	×	×	×	×	×
**DB2**	×	—	×	√	√	√	√	√	√	√	√
**DB3**	×	×	—	0	0	0	0	0	0	0	0
**DB4**	√	√	0	—	√	√	√	√	√	√	√
**DB5**	√	√	0	√	—	√	√	√	√	×	√
**DB6**	√	√	0	√	√	—	√	√	√	√	√
**DB7**	×	√	0	√	√	√	—	×	×	×	×
**DB8**	×	√	0	√	√	√	×	—	×	×	×
**DB9**	×	√	0	√	√	√	×	×	—	×	×
**DB10**	×	√	0	√	×	√	×	×	×	—	×
**DB11**	×	√	0	√	√	√	×	×	×	×	—

(Note: ”√” means compatible,”×” indicates incompatibility, “0” means it is not necessary to consider compatibility).

**Table 2 sensors-22-02373-t002:** Structure parameter table of CNN with an image resolution of 180 × 320.

Convolution Blocks	Filter Size	Number of Filters	Feature Image Dimension
1	3 × 3	8	180 × 320 × 8
2	3 × 3	16	90 × 160 × 16
3	3 × 3	32	45 × 80 × 32
**Fully Connected Layer**	1	**Number of Hidden Units**	115,200

**Table 3 sensors-22-02373-t003:** Structure parameter table of CNN with an image resolution of 360 × 640.

Convolution Blocks	Filter Size	Number of Filters	Feature Image Dimension
1	3 × 3	8	360 × 640 × 8
2	3 × 3	16	180 × 320 × 16
3	3 × 3	32	90 × 160 × 32
**Fully Connected Layer**	1	**Number of Hidden Units**	460,800

**Table 4 sensors-22-02373-t004:** Comparison of cascaded models and non-cascaded model for daytime data sets.

	Hand Posture	Not Looking Ahead	Smoking	Calling the Phone	Behavior Posture	Non-Cascaded Model	Cascaded Models
**Accuracy**	99.3%	98.94%	98.59%	99.3%	99.35%	97.83%	98.68%
**Speed**	91 ms	93 ms	90 ms	89 ms	89 ms	452 ms	405 ms

**Table 5 sensors-22-02373-t005:** Comparison of cascaded models and non-cascaded model for night data sets.

	Hand Posture	Not Looking Ahead	Smoking	Phone Calling	Behavior Posture	Non-Cascaded Model	Cascaded Models
**Accuracy**	98.14%	99.78%	98.6%	98.6%	99.77%	97.48%	98.03%
**Speed**	82 ms	80 ms	79 ms	81 ms	81 ms	403 ms	362 ms

## Data Availability

Not applicable.

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
