# Peer review of "A Proactive Recognition System for Detecting Commercial Vehicle Driver’s Distracted Behavior"

_sensors, 2022, doi:10.3390/s22062373_

Round 1
Reviewer 1 Report
This paper proposes a methodology to study the recognition of driver's distracted behaviour based on computer vision and convolutional neural networks. The method tries to enlarge the the area of recognition and also the objects and scenarious to be considered in the models. The study proposes five CNN sub models aimed at recognizing different posture categories and compares the results of using these sub models in a non-cascaded or cascaded set-up.
In order for the reader to better understand the presented study, authors should take into consideration the following comments and suggestions:
- First sentence in paragraph 2, page 10 is confusing, it must be rephrased.
- Table1 header must be enlarged to show postures types correctly (i.e. DB10 and DB11).
- In last paragraph, page 11, is mentioned fig. 5a, but it should be fig. 8a
- In first paragraph in section 5.1, is mentioned fig 7a, but it should be fig. 10.a. See also subsequent numbering errors of figures. Use also same format for figure reference in text (either bold or normal)
- Section 5. Why were selected different percentages for daytime/night training/test sets? 24.2% vs. 23.7%
- In figures 12-14 the 100% accuracy is reached earlier than 10 iterations. Comments?
- Table 4 and 5. How was computed the accuracy of the non-cascaded model?
- Correct word Acknowledgement
- Two other distracted behaviours that can result in a reckless driving or speed unsafe for conditions are talking with passengers and texting. Did authors take into consideration also these types of distracted behaviour or do they plan to do it in future studies?
- Considering the posture labeling, it could be studied if a different labeling could provide an improvement of the recognition algorithm (ex. more dangerous distracted behaviours be given higher ranks in the order, keeping at the same time the selected grouping).
Reviewer 2 Report
The paper presents a very focus study of associating a certain class of driver's posture to distracted behavior. The paper setup such recognition based on 5 ad-hoc significant characteristics which are well-motivated. Then they define eleven features associated with the driver's posture and activities based on the collected video samples (which include both day and night time driving). A deep learning architecture is proposed where the authors have experimented with various combinations for arranging and labeling the feature vector used in the CNN architecture. They have shown various convincing experimental results based on their training and testing samples for two types of CNN architecture.
Overall, the paper is a complete exposition starting with a problem definition, detailed breakdowns, labeling and deep-learning architecture (such as Cascaded architecture) (with convincing experimental results). However, it is not yet clear how the proposed posture detection can be coupled with the traffic and road conditions. For example, in cruise controlled condition and empty road, prolong duration of time for controlling the dashboard or taking things may not be considered a distraction which can be a safety hazard.
One of the main safety issues is the driver's drowsiness which would be associated with the head area (shown in Figure 6). It is not clear how such event can be detected for the proposed camera location.
It would also make more sense if a level of distraction can be proposed with the warning sign indicating dangerous condition.
